# Normal male fertility in a mouse model of KPNA2 deficiency

**Franziska Rother**[1,2]*, **Dalia Abu Hweidi**[1], **Enno Hartmann**[2], **Michael Bader**[1,2,3,4]

**1** Max Delbrück Center for Molecular Medicine, Berlin, Germany, **2** Institute for Biology, University of Lübeck, Lübeck, Germany, **3** Charité - Universitätsmedizin Berlin, Corporate Member of Freie Universität Berlin and Humboldt-Universität zu Berlin, Berlin, Germany, **4** DZHK (German Center for Cardiovascular Research), Partner Site Berlin, Berlin, Germany

* franziska.rother@mdc-berlin.de

**Data Availability Statement:** All relevant data are within the manuscript and its Supporting Information files.

**Funding:** The author(s) received no specific funding for this work.

## Abstract

The nuclear transport of proteins is mediated by karyopherins and has been implicated to be crucial for germ cell and embryonic development. Deletion of distinct members of the karyopherin alpha family has been shown to cause male and female infertility in mice. Using a genetrap approach, we established mice deficient for KPNA2 (KPNA2 KO) and investigated the role of this protein in male germ cell development and fertility. Breeding of male KPNA2 KO mice leads to healthy offsprings in all cases albeit the absence of KPNA2 resulted in a reduction in sperm number by 60%. Analyses of the KPNA2 expression in wild-type mice revealed a strong KPNA2 presence in meiotic germ cells of all stages while a rapid decline is found in round spermatids. The high KPNA2 expression throughout all meiotic stages of sperm development suggests a possible function of KPNA2 during this phase, hence in its absence the spermatogenesis is not completely blocked. In KPNA2 KO mice, a higher portion of sperms presented with morphological abnormalities in the head and neck region, but a severe spermiogenesis defect was not found. Thus, we conclude that the function of KPNA2 in round spermatids is dispensable, as our mice do not show any signs of infertility. Our data provide evidence that KPNA2 is not crucial for male germ cell development and fertility.

## Introduction

The trafficking of proteins between nucleus and cytoplasm is an essential mechanism to serve the particular needs of a developing cell. It enables compartmentalization of the cell, yet ensures, that proteins become available in the nucleus whenever needed. The classical mechanism of nuclear import involves karyopherin α (KPNA, also named importin α) and karyopherin β1 (KPNB1, also named importin β1). When KPNA binds to distinct nuclear localization sequences (NLS) present in target cargo proteins, it also makes contact with KPNB1, which in turn mediates the contact with the nuclear pore [1]. In the classical pathway, the three proteins—KPNA, KPNB1 and cargo—together enter the nucleus where the trimeric complex dissociates and the cargo protein is released to fulfill its function. The family of

**Competing interests:** The authors have declared that no competing interests exist.

karyopherins consists of more than 20 proteins, categorized into KPNA and KPNB; however, only one KPNB, namely KPNB1, participates in the classical nucleocytoplasmic transport pathway. On the other hand, six different KPNA paralogues have been found in mice and seven in rats and humans [2–4]. These paralogues show clear differences, not only in their cellular and organ specific expression but also in their subcellular localization. Moreover, different KPNA paralogues display distinct binding specificities towards potential substrates, indicating that they play diverse roles in various cellular pathways. Additionally, KPNAs have been implicated in cellular processes other than nucleocytoplasmic transport, like nuclear envelope formation or spindle assembly [5,6]. The differential expression of KPNA paralogues in combination with their specificity for distinct substrates can lead to severe effects if one paralogue is missing, because other paralogues might not be able to compensate for the loss [7–13].

The function of the male reproductive system depends on an undisturbed spermatogenesis within the seminiferous epithelium of the testis. During this process, spermatogonia undergo mitosis and differentiate into primary spermatocytes, which process through preleptotene, leptotene, zygotene, pachytene, and diplotene stages of meiosis I to generate secondary spermatocytes. Subsequently, secondary spermatocytes enter the second meiotic division resulting in round spermatids. The haploid round spermatids undergo dramatic morphological changes and finally differentiate into mature spermatozoa [14]. This ordered sequence of division and differentiation steps repeatedly requires changes in gene expression, coordinated by a plethora of transcription and other factors found in the testis [15,16]. The expression of members of the KPNA family in the male testis has been investigated on mRNA- and, to a lesser extent, on protein levels, revealing distinct expression patterns for different KPNA paralogues, which indicates that germ cell differentiation and maturation could be controlled by nucleocytoplasmic transport events [17–19]. Previous data suggest, that KPNA2 is expressed at very high levels in the testis and could be critical for male fertility [20,21]. We and others have recently shown, that loss of KPNA2 is absolutely essential for early embryonic development in mice due to a maternal effect leaving oocytes derived from KPNA2-deficient mice unable to develop past the 2cell embryonic stage [22]. This results in female infertility in KPNA2 deficient mice. The recent work by Navarrete-López et al. using a model of KPNA2 knockout (KO) mice has suggested that KPNA2 could also play a crucial role for male fertility [21]. In the present study we now investigate the role of KPNA2 for male germ cell development and fertility in mice.

## Materials and methods

### Ethics statement

All studies involving animals were performed according to the current national guidelines for the humane use of laboratory animals and reported to the Landesamt für Gesundheit und Soziales (LAGESO) Berlin.

### Generation of KPNA2 KO mice

The generation of KPNA2 KO mice has been described recently [23]. Briefly, ES cells with a gene trap mutation in the KPNA2 gene (clone XS0061, ES cell line E14Tg2a.4 from background 129P2/Ola) were purchased from MMRRC at University of California, Davis, and directly used for blastocyst injection. Embryos were recovered by flushing the uterine horns with M2 medium (Sigma) from superovulated (7.5 IU PMSG on day 1 at 1pm; 7.5 IU hCG on day 3 at 12am) 23–25 days old C57BL/6N female mice at 3.5dpc. 15–17 ES cells were injected per blastocyst and embryos were transferred into uterine horns of 2.5dpc anesthetized (ketamine 100 mg / kg and xylazine 10 mg / kg body weight) pseudopregnant NMRI females (10–

12 embryos per recipient). Germline chimeras were bred with C57BL/6N mice and the resulting offspring were backcrossed for at least 8 generations to C57BL/6N background before colonies were maintained by breeding the resulting heterozygous mice. For the genotyping, a three-primer PCR was developed using the following oligonucleotides: 5'-AGT TCT GAT GGG CAA CCA AG-3'; 5'-CTT AAT GTG GGC AGC ACC ATC-3'; 5'-CCT CCG CAA ACT CCT ATT TC-3'; resulting in a wild-type (WT) fragment of 794bp and a KO fragment of 597bp.

## Husbandry and sacrifice of animals

Animals were bred according to the current regulations of the LAGESO, which allow only breeding of the number of animals that will later be used in experiments. Animals were kept in groups in cages with enhanced environment and free access to food and water and visited daily by animal caretakers. For the retrieval of mouse tissues, animals were sacrificed by cervical dislocation without anesthesia by an experienced scientist.

## RNA isolation and RT-PCR

Total RNA was isolated from mouse tissues by use of the TRIZOL method. Briefly, the material was homogenized in 1 ml Trizol reagent, 0.2 ml chloroform was added and after centrifugation, the colorless upper phase was transferred to a clean tube. Then 0.5 ml isopropyl alcohol was added, and after careful washing with ethanol, the RNA was air dried and dissolved in RNAse free water. The synthesis of cDNA was achieved by incubation in a 20 μl reaction mixture containing 200 U of MMLV Reverse Transcriptase (Promega) and 500 ng of random primer (Roche) at 25°C for 15 minutes, followed by 60 minutes incubation at 37°C and inactivation for 15 minutes at 70°C. The cDNA was diluted to 5 ng/μl. For PCR, a 25 μl reaction mixture consisted of 10 ng of the cDNA solution, 50 ng of each primer, 5 μM dNTP, and 1 IU Taq DNA polymerase (NEB). For detection of KPNA2 mRNA a forward primer in exon 1 (5'-GAA GGG TAG CAG ACG TTT CC-3') and reverse primer in exon 4 (5'-AAC AAT GTC CTC AAC AGA CC-3') were used, while for detection of the modified mRNA the forward primer in exon 1 was combined with a reverse primer in the BetaGeo sequence (5'-GTT TTC CCA GTC ACG ACG TTG-3').

## SDS PAGE and western blot

For western blot of mouse testis, 30 μg of tissue protein extracts were loaded on a 10% SDS gel. After transfer of proteins, the PVDF membrane was blocked by Odyssey blocking solution (LiCor, Bad Homburg, Germany) and subsequently incubated with primary antibodies at 4°C overnight (C-term. antibody: anti-KPNA2 (goat), Origene TA302917, 1:8,000; N-term. antibody: anti-KPNA2 (rabbit), selfmade; peptide MSTNENANLPAARL, 1:1,000; anti-GAPDH (rabbit), Cell Signaling 2118, 1:1,000). On the next day, the membrane was incubated with an IRDye- coupled secondary antibody for 1 h at room temperature (IRDye 800 donkey anti-rabbit LiCor (926–32213) and IRDye 800 donkey anti-goat LiCor (926–32214), both 1:10,000) and detection was performed using the Odyssey Infrared Scanner (LiCor, Bad Homburg, Germany).

## Immunohistochemistry

For histological analyses, testes were fixed in neutral buffered 4% formalin. After fixation, tissues were dehydrated in increasing concentrations of ethanol, embedded in paraffin wax, and sectioned at a thickness of 5 μm. Sections were deparaffinised, rehydrated and underwent

antigen retrieval using citrate buffer pH6 for 20 minutes. The sections were then treated with 10% normal donkey serum for 1h at room temperature and subsequently incubated with KPNA2 antibody (Origene TA302917 1:100) overnight at 4˚C. On the next day, sections were washed with PBS, incubated with secondary antibody (donkey anti goat Cy3 Jackson ImmunoResearch 705-165-147 1:500), washed and counterstained for 2h with peanut agglutinin (PNA) coupled to Alexa 488 to visualize the acrosome. Finally, after washing again, the sections were embedded in mounting medium containing DAPI (Vectashield, Vector Laboratories/Biozol, Germany). Images of stained tissue sections were taken using a fluorescence microscope (Keyence, Bioreva BZ-9000, Germany) or a confocal fluorescence microscope (Leica TCS SPE).

### Breeding of KPNA2 males

KPNA2 WT, HET and KO males were bred for 3–8 months with WT females (1:2, 8–13 males of each genotype) to assess fertility and litter size. Whenever the breeding resulted in more than one litter, the mean litter size of the respective male was calculated. Thus, each data point corresponds to a single male individual.

### Epididymal sperm count

Sperm count was performed as described previously [24,25]. Briefly, one caudal epididymis was minced in 1 ml of PBS. Sperms were allowed to disperse into solution by incubating for 10 min at 37˚C temperature. An aliquot of the sperm/saline mixture was then counted in a hemocytometer. The hemocytometer count was multiplied by appropriate volume and dilution factors to give a total cauda epididymal sperm count.

### Sperm morphology assessment

For the assessment of sperm morphology, 6μl sperm suspension was spread onto a slide. The staining was performed using the Spermac Kit (Minitube) according to manufacturer's instructions. 200 sperms per mouse were analysed under a light microscope (Keyence, Bioreva BZ-9000, Germany), classifying the abnormalities in the following categories: normal morphology, head and neck defects, midpiece defects, tail defects, multiple defects.

### Statistical analyses

Statistical analysis was performed with Prism7 (GraphPad). Results are presented as means ± s.e.m. Significance was determined by using ANOVA (where 3 groups were compared), for multiple comparisons, Dunnett's multiple comparisons test was applied. In analyses comparing 2 groups, the unpaired two-tailed Student's t test was used. Significance was assumed for $p < 0.05$ (*, $p < 0.05$; **, $p < 0.01$; ***, $p < 0.001$; ****, $p < 0.0001$; n.s., not significant).

## Results

### KPNA2-deficient male mice are fertile

We generated KPNA2 knockout (KO) mice using a gene trap approach. These mice carry a genetrap cassette in intron 3/4 of the KPNA2 gene leading to the transcription of a modified mRNA from the KPNA2 promoter with subsequent formation of a fusion protein containing the importin beta binding domain of KPNA2 and a β-galactosidase-neomycin fusion protein (Fig 1A) [23]. RT-PCR analysis confirmed the absence of KPNA2 mRNA in the testis of KO mice, while the modified mRNA is formed (Fig 1B and S2 File). To confirm that KPNA2

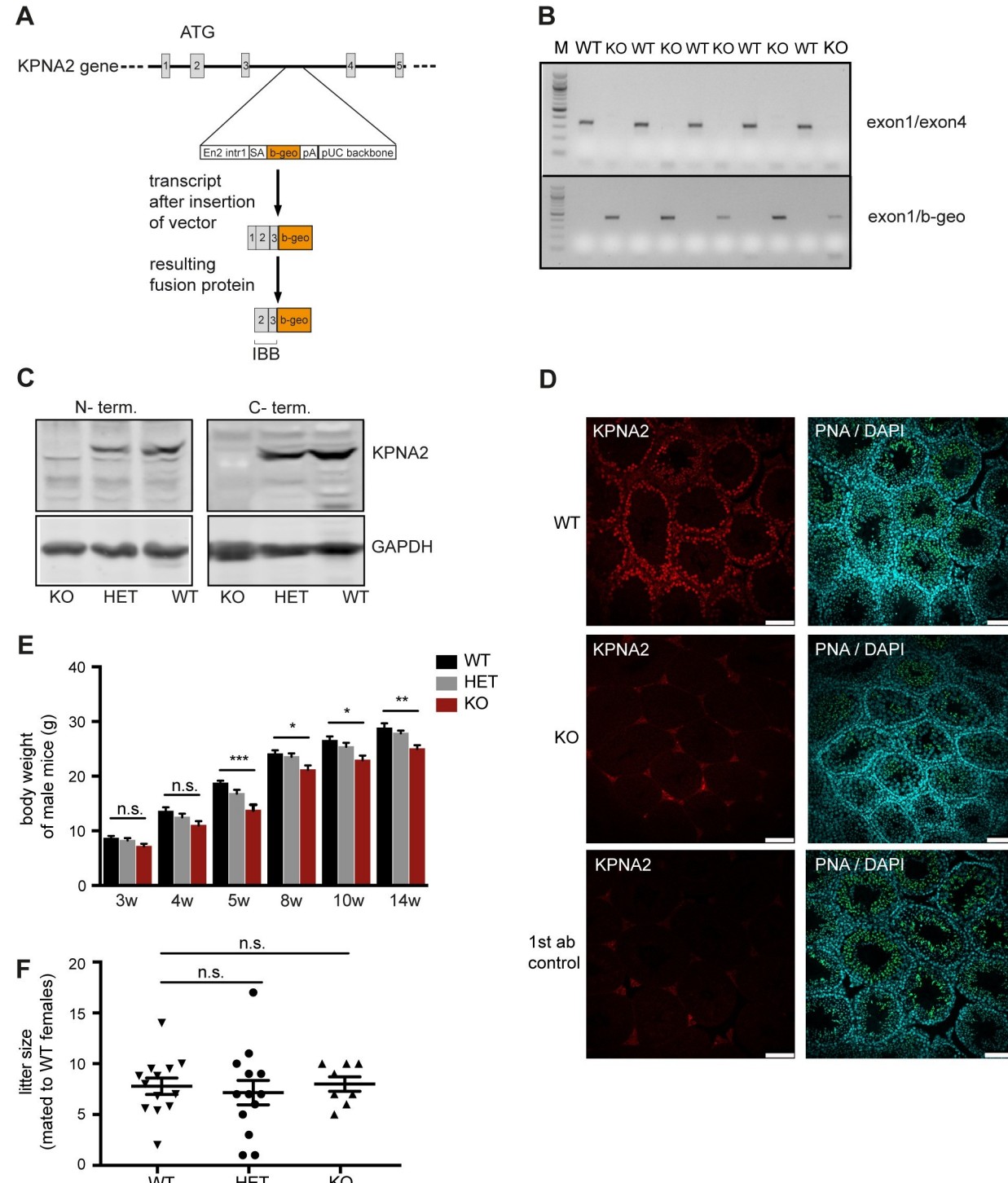

**Fig 1. Generation and characterization of KPNA2 KO mice.** (A) A genetrap cassette containing a β-galactosidase and neomycin resistance gene (β-geo) was introduced into intron 3/4 of the KPNA2 gene. This results in aborted transcription of KPNA2 with formation of a fusion protein containing the importin beta binding domain of KPNA2 and β-geo fusion protein. (B) RT-PCR of KPNA2 in the testis of various wild-type (WT) and KPNA2-knockout (KO) mice using primers spanning from exon1 into exon4 and from exon1 into the β-geo sequence. M: 100bp marker. (C) Western Blot for KPNA2 in the testis of WT, heterozygous (HET) and KPNA2 KO mice using N-terminal (left panel) and C-terminal (right panel) antibodies. (D) Immunofluorescence staining of KPNA2 (red) in the testis of WT and KPNA2 KO mice counterstained with PNA for visualization of the acrosome (green) and DAPI (blue). The no 1st antibody control shows the unspecific background of the secondary antibody. Scale bars 100 μm. (E) Body weight of male KPNA2 KO, HET and WT mice at various ages (number of mice per group: 10–19). (F) Litter size of KPNA2 deficient males mated with WT females (number of males per group: 8–13).

protein is deleted in the testis, we performed a western blot analysis with C-terminal and N-terminal KPNA2 antibodies which both revealed the absence of the protein while it is still detectable in KPNA2 heterozygous (HET) mice (Fig 1C and S2 File). Immunohistochemistry of the testis confirmed the absence of KPNA2 in KO mice (Fig 1D). As already seen for female mice [23], the male KPNA2 KO mice display a reduced body weight compared to WT littermates (Fig 1E and S1 File). Interestingly, and in contrast to females, KPNA2 KO males proved to be fertile and no differences in litter size could be found when comparing the three groups (mean litter size in WT: 7.8, in HET: 8.4, in KO: 8.0, Fig 1F and S1 File).

## KPNA2 is highly expressed in meiotic spermatocytes

To further evaluate the presence of KPNA2 in WT mouse testis, we performed a detailed analysis of KPNA2 protein expression and localization throughout all stages of spermatogenesis (Fig 2A and 2B). We found a nuclear KPNA2 expression in spermatocytes starting in preleptotene stage of meiotic prophase which raised continuously during the subsequent leptotene, zygotene and pachytene stages. Showing its highest expression in late pachytene stage and in diakinesis stage spermatocytes, the KPNA2 levels subsequently started to decrease in metaphase stage. In secondary spermatocytes KPNA2 was found in the nucleus and cytoplasm, while early round spermatids again displayed a nuclear, albeit weaker, KPNA2 staining pattern. After step 1, the nuclear expression of KPNA2 rapidly decreased in subsequent steps of sperm maturation. No KPNA2 could be found in Sertoli cells and Leydig cells, the latter displaying a high background signal (Fig 1D).

## Reduced sperm count and aberrant sperm morphology in KPNA2-deficient mice

The pronounced expression of KPNA2 in meiosis stage spermatocytes indicates a function of KPNA2 during this developmental stage. To understand, in which way and how strongly the absence of KPNA2 during meiosis effects the formation of mature sperms, we analyzed sperms retrieved from the epididymis of adult mice. We found a pronounced reduction in sperm number of about 60% in KPNA2 KO mice while the sperm count of HET mice was unchanged compared to WT (mean epididymal sperm count in WT: $11.6*10^6$, in HET: $10.2*10^6$, in KO: $4.7*10^6$, Fig 3A and S1 File). Next, we investigated the sperm morphology of epididymal sperms. We found a significant increase in head and neck defects in KPNA2 KO sperms compared to WT, suggesting a role for KPNA2 in formation of sperm head and neck, while tail morphology and midpiece were not affected (WT: 55.4% normal sperms, 19.3% head/neck abnormalities; KO: 38.4% normal sperms, 42.5% head/neck abnormalities; Fig 3B and 3C and S1 File).

## Discussion

The sequence of events in germ cells that lead to the formation of mature sperms able to fertilize an oocyte consists of multiple steps including two meiotic divisions, a massive morphological transformation and even the complete unpacking and repacking of the sperm DNA, when histones are replaced by protamines. Consequently, individual phases during sperm development have different requirements regarding gene and protein expression, trafficking of proteins—e.g. transcription factors -, or prevention of aggregation of proteins at a certain localization. In this light it does not come as surprise that nucleocytoplasmic transport events have been implicated to play an important role during germ cell differentiation and maturation [18,26–28]. Earlier studies have revealed that different KPNA paralogues show unique expression patterns in the testis [13,17–20]. With respect to protein expression in murine

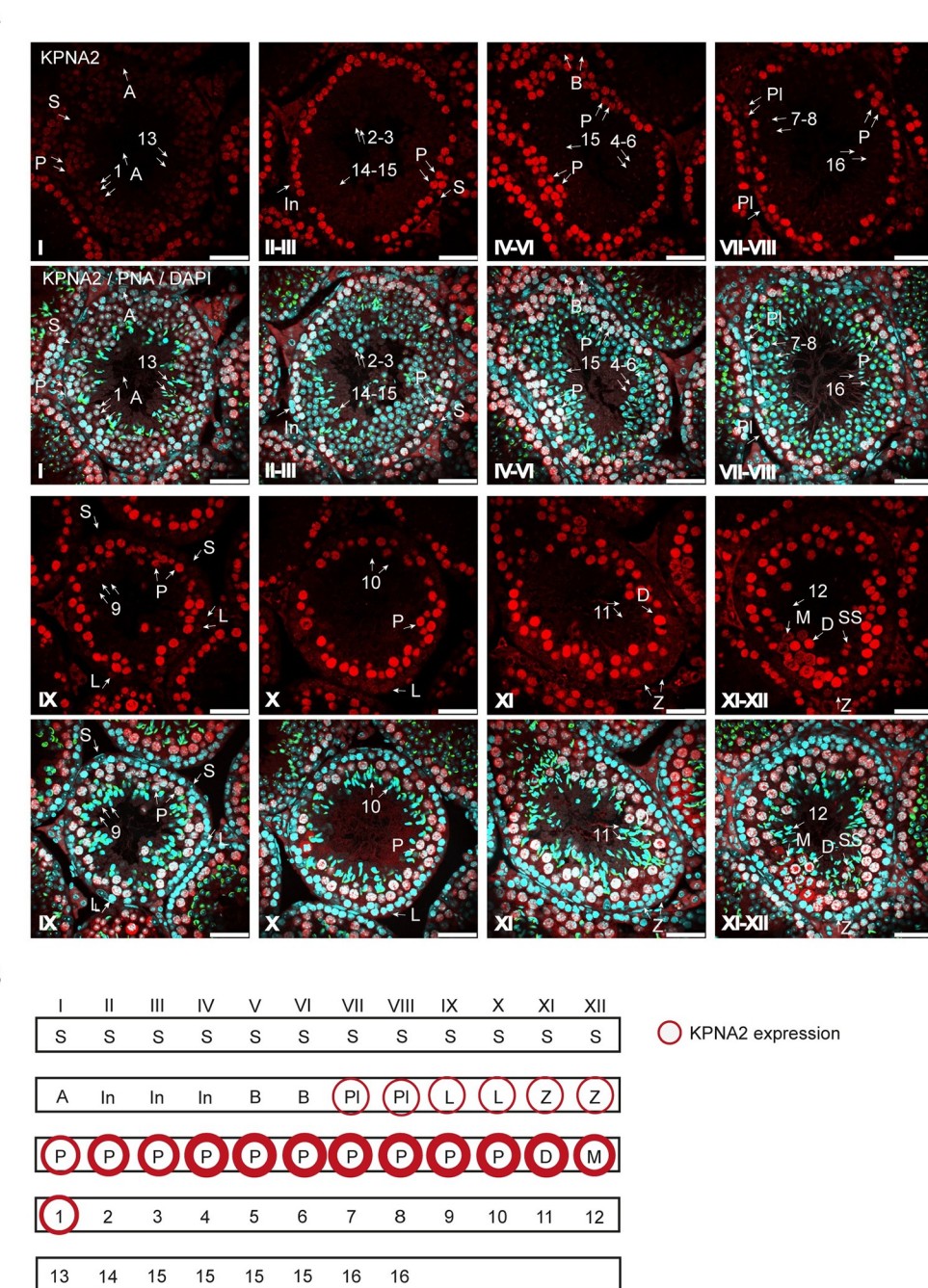

**Fig 2. Expression and localization of KPNA2 in the murine testis.** (A) Immunohistochemistry of testis sections of adult mice (12–16 weeks) using an antibody against the C-terminus of KPNA2 (red), counterstained with DAPI (blue) and PNA, labelling the acrosome (green). Roman numbers mark tubular stages. Scale bars: 50 μm. (B) Schematic image of KPNA2 expressing cell types in the mouse testis. S: Sertoli cell; A: Type A spermatogonium; In: Intermediate spermatogonium; B: Type B spermatogonium; Pl: Preleptotene spermatocyte; L: Leptotene spermatocyte; Z: Zygotene spermatocyte; P: Pachytene spermatocyte; D: Diakinesis spermatocyte; M: Metaphase of meiosis I, SS: Secondary spermatocyte. Arabic numbers mark developmental steps of haploid round and elongating spermatids (1–16).

testis, it has been shown that KPNA4 localizes to nuclei of Sertoli cells, pachytene spermatocytes, and round spermatids step 7–8, while KPNA3 is ubiquitously expressed in the cytoplasm of Sertoli cells, mitotic and meiotic spermatocytes as well as round and elongating spermatids

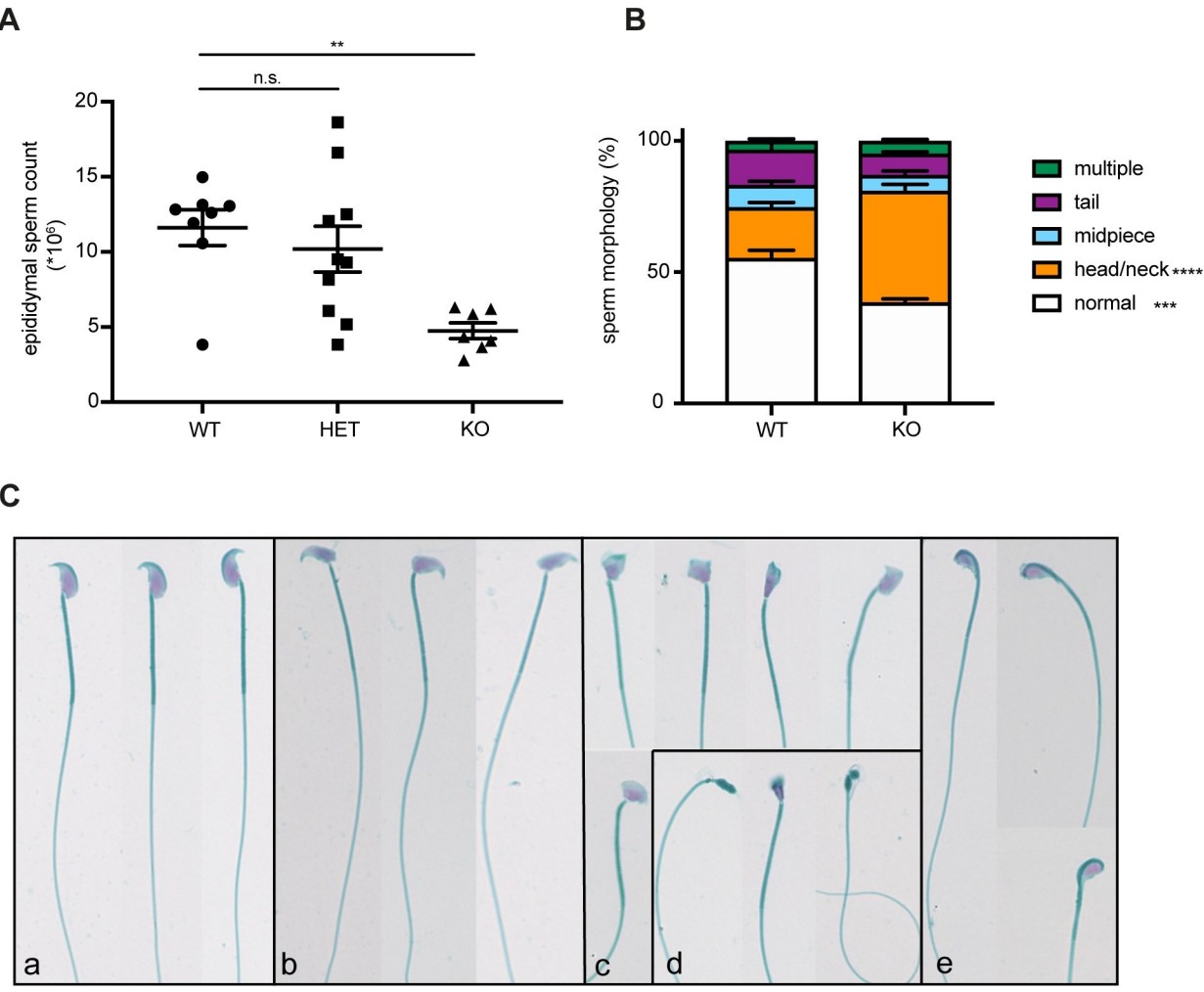

**Fig 3. Aberrant sperm morphology in KPNA2 KO mice.** (A) Epididymal sperm count (n = 7–10 mice per group, age 12–16 weeks). (B) Quantification of morphological sperm abnormalities (n = 3 mice per group, age 12–16 weeks). (C) Representative images of epididymal sperms from WT (a), and KPNA2 KO (b-e) mice. The majority of abnormal sperms displayed deformations of the sperm head with a divergent head-to-neck angle (b), deformed head (c), absent or severely disrupted head formation (d) or head bended-down (e).

[29]. The protein expression of KPNA6 was found to be restricted to elongating spermatids [13].

In the present study, we used a KPNA2-deficient mouse model to investigate the role of KPNA2 during spermatogenesis and postmeiotic spermatid maturation. In contrast to a recently published study by Navarrete-López et al., in which KPNA2 was found to be essential for male germ cell formation and murine fertility [21], we did not observe such an effect. Our KPNA2 KO mice were generated using a genetrap method, completely eliminating mRNA and protein expression of KPNA2 and we could confirm the growth retardation of KPNA2 KO mice found by Navarrete-López et al.[21]. However, our male KPNA2 KO mice are fertile and litter size is not different between WT, HET and KO males that have been mated to WT females. We found KPNA2 protein to be highly expressed during all stages of meiosis and in secondary spermatocytes, where it shows a strong nuclear localization, suggesting that the major role of KPNA2 occurs during these stages of germ cell development. Interestingly, and in contrast to the work executed by Navarrete-López et al. [21], the KPNA2 expression in

round spermatids declined rapidly, with only early round spermatids revealing a KPNA2 signal. This let us conclude, that KPNA2 could be dispensable for spermiogenesis in our KO mice. The sperm counts retrieved from epididymal sperms support this hypothesis: we found a 60% reduction in sperm number and among these sperms a higher rate of abnormal morphology; but around 38% of KPNA2 KO sperms did not show any abnormalities. While in WT mice 55.4% of $11.6*10^6$ epididymal sperms show a normal morphology (around $6.4*10^6$ sperms), in KO there are still 38.4% of $4.7*10^6$ epididymal sperms normal (around $1,8*10^6$ in total). These remaining KO sperms with a normal morphology seem to be sufficient for fertilizing single oocytes, as it proved to be the case in our mouse model. Importantly, the morphological analysis of sperms confirmed the findings by Navarrete-López et al. [21]: head and neck defects were more abundant in KPNA2 KO compared to WT in our model.

The nuclear import of transcription factors after successfully completed meiosis is a crucial event during spermiogenesis: the newly formed haploid round spermatids massively enhance their transcription to produce mRNA and proteins needed for the following maturation, morphological reorganization and DNA repacking. A disturbed nuclear import of transcription factors as a consequence of KPNA-deficiency should lead to severe effects in resulting sperms. Such a phenotype has been observed in KPNA6 KO mice, where the translocation of the transcription factor BRWD1 and the expression of transition nuclear proteins and protamines was markedly reduced, which resulted in a total infertility and reduced sperm count < 3% compared to WT [13]. In line with these observations, and in contrast to KPNA2, KPNA6 is highly expressed in round and elongating spermatids, while no KPNA6 expression is found in meiotic germ cells. On the other hand, the strong expression of KPNA2 in meiotic germ cells and its rapid decrease in early round spermatids supports a central role during meiosis rather than a transport defect in late sperms, as it has been seen by Navarrete-López et al. [21]. In this light, it will be interesting to further investigate the function of KPNA2 during meiosis and to understand why, although highly expressed during this stage, KPNA2 is not essential. One hypothesis could be that another KPNA paralogue can compensate for the loss.

Different phenotypes in KO mice sometimes can be attributed to different genetic backgrounds. The mice used in the present study were backcrossed to C57BL/6N background, while Navarrete-López et al. used mice with a mixed C57BL/6 and CBA background [21]. Interestingly, a recently published study by Wang and colleagues confirmed our data, that KPNA2 is not essential for male fertility [22]. The authors of this study had used a CRISPR/Cas9 approach to eliminate KPNA2 and bred the mice on ICR background which is known for high reproductivity. The models are genetically divers, which could be a reason for observed differences. However, both models–our fertile KPNA2 KO mouse and the infertile KPNA2 KO mouse created by Navarrete-López et al.—are consistent regarding the growth retardation, average litter size and genotypes of offspring, which do not display Mendelian ratios [23]. We conclude, that despite different backgrounds, these phenotypes are evident and therefore strongly related to the missing KPNA2. On the other hand, we present here a mouse model of KPNA2 deficiency with normal fertility in male mice, proving, that KPNA2 is not essential for germ cell development and maturation.

## Supporting information

**S1 File. Animal phenotype raw data.** Body weight, litter size, epididymal sperm count and morphology of sperms of KPNA2 KO mice are provided as original data.
(XLSX)

**S2 File. Raw images of gels and blots.**
(PDF)

## Acknowledgments

The authors wish to thank Anne Hahmann, Andrea Rodak, Madeleine Skorna-Nussbeck for technical assistance. We thank Raisa Brito Santos for her expertise concerning sperm morphology. We also thank the Advanced Light Microscopy technology platform of the MDC for technical support.

## Author Contributions

**Conceptualization:** Michael Bader.

**Data curation:** Franziska Rother.

**Formal analysis:** Franziska Rother.

**Funding acquisition:** Enno Hartmann, Michael Bader.

**Investigation:** Franziska Rother, Dalia Abu Hweidi.

**Methodology:** Franziska Rother.

**Project administration:** Franziska Rother, Michael Bader.

**Resources:** Enno Hartmann, Michael Bader.

**Software:** Michael Bader.

**Supervision:** Enno Hartmann, Michael Bader.

**Validation:** Franziska Rother.

**Visualization:** Franziska Rother.

**Writing – original draft:** Franziska Rother.

**Writing – review & editing:** Enno Hartmann, Michael Bader.

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
