## [Decision Letter · Decision Letter 0]

9 Aug 2024

PONE-D-24-18326Normal male fertility in a mouse model of KPNA2 deficiencyPLOS ONE

Dear Dr. Rother,

Thank you for submitting your manuscript to PLOS ONE. After careful consideration, we feel that it has merit but does not fully meet PLOS ONE’s publication criteria as it currently stands. Therefore, we invite you to submit a revised version of the manuscript that addresses the points raised during the review process.

We look forward to receiving your revised manuscript.

Kind regards,

Xuejiang Guo, Ph.D.

Academic Editor

PLOS ONE

Journal Requirements:

2. To comply with PLOS ONE submissions requirements, in your Methods section, please provide additional information regarding the experiments involving animals and ensure you have included details on (1) methods of sacrifice, and (2) efforts to alleviate suffering.

Reviewers' comments:

Reviewer's Responses to Questions

**Comments to the Author**

1. Is the manuscript technically sound, and do the data support the conclusions?

Reviewer #1: Yes

Reviewer #2: Yes

2. Has the statistical analysis been performed appropriately and rigorously? 

Reviewer #1: Yes

Reviewer #2: Yes

3. Have the authors made all data underlying the findings in their manuscript fully available?

Reviewer #1: No

Reviewer #2: Yes

4. Is the manuscript presented in an intelligible fashion and written in standard English?

Reviewer #1: Yes

Reviewer #2: Yes

5. Review Comments to the Author

Reviewer #1: In the present manuscript, the authors were interested in deciphering the role of one of the karyopherin alpha family proteins, KPNA2, implicated in protein nuclear transport, in male germ cell development and fertility in mice. Authors have previously shown that KPNA2 is essential for early embryonic development in mice, which deficiency results in female infertility. The present work and the previous one were based in a genetically modified mouse model, KPNA2 KO, produced by gene-trapping in embryonic stem cells. The main conclusion is that KPNA2 is dispensable for spermiogenesis, and its absence does not affect male fertility. However, some changes were observed comparing to WT littermate males: sperm counting was significantly reduced, head/neck sperm abnormalities were seen, and males showed a reduced body weight.

REVIEWER COMMENTS:

Below, I have some suggestions and comments with the aim to make the M&M section more complete and the Results section clearer. I hope the authors can take them into consideration. Manuscript should have had number lines to make correction easier.

MATERIALS AND METHODS

Generation of KPNA2 KO mice. I suggest the authors should refer the generation of this KO line to the work in which it was first published (Rother, F. et al, 2024. Karyopherin α2 is a maternal effect gene required for early embryonic development and female fertility in mice. FASEB Journal. 2024;38:e23623) because this model was already used by them in a previous work. Although Animal Ethics statement was uploaded on the submission process, I would suggest to add this information to the M&M section.

Regarding blastocyst injection with ESC, there are some missing details. I suggest the authors could add the strain of the ESC clone purchased at MMRRC and details of the following procedures: superovulation of blastocyst donor females (hormones, times), blastocyst collection and embryo transfer (surgical or non-surgical technique was used? Reagents and doses of anesthesia and analgesia if used). Please detail the mouse strain/substrain of the blastocyst donor females and recipient females.

The name of the strain C57BL/6 is misspelled, correct the “L”. In the Discussion section, it appears that the C57BL/6N strain was used, please use the name of this substrain in the M&M section.

Details regarding body weight analysis are completely missing. How many animals of each age and genotype were evaluated?

The procedures regarding breeding of KPNA2 KO mice to evaluate fertility should be added. How many matings were tested, type of mating (1 female x 1 male; 2 females x 1 male; etc), duration of the mating (weeks, months).

Following PLOS One policy, all data underlying the findings in the manuscript should be fully available. I suggest the authors to report all the data in tables as supplemental files: Body weight of male mice, litter size, epididymal sperm count and sperm morphology.

RESULTS

For a clearer understanding of the results, the mean numbers of the following parameters, for each tested group, would be added to the main text: litter size; epididymal sperm counts; normal and head/neck abnormal sperm percentages. It complements the information showed in the graphs.

DISCUSSION

Please, avoid the term “fathers” by explaining the different matings evaluated.

I suggest changing the following phrase: “Differently observed phenotypes in KO mice” to “Different phenotypes in KO mice…”

Please, correct the B6 strain name (C57BL/6).

The following phrase should be corrected to be understandable: This leaves around 14% of sperms (compared to WT) with a normal morphology, which is likely to be sufficient for fertilizing single oocytes, as it proved to be the case in our mouse model. Where does the 14% came out?

In Wang et al. 2023 is concluded that KPNA2 in mice could play the role of KPNA7 in humans, which deficiency has been related to preimplantation embryo arrest. Deficiency of KPNA7 in male humans, has been studied/reported? If so, a brief comment could be added to the discussion section.

FIGURE LEGENDES

Correct “und” by “and” in Figure B) RT-PCR of KPNA2

REFERENCE LIST

I suggest changing reference #23 to the peer-reviewed one (The FASEB Journal. 2024;38:e23623).

Reviewer #2: The MS submitted by Rother et al. describes in an elegant way the characterization of a KPNA2 KO mice regarding the fertility in males. This protein has been already studied in female mice leading to a deffect in embryo development, and in KO male mice leading to infertlity, in this case generated by another research group. In the present MS the males did not show signs of unfertility, although sperm count and abnormalities were similar to the previous reported model. The main difference can be attributed to the genetic background of the mice, something that accounts for multiple difference among mice strain and subtrains. Language is very clear and the experiments have been carried out appropiately with a proper discussion.

In my opinion, the MS can be published in its present form.

6. PLOS authors have the option to publish the peer review history of their article (what does this mean?). If published, this will include your full peer review and any attached files.

Reviewer #1: No

Reviewer #2: **Yes: **Martina Crispo

---

## [Author Response · Author response to Decision Letter 0]

11 Sep 2024

Thank you for the valuable comments and suggestions. We have carefully fulfilled the required changes and attached a "response to reviewers" document with point by point comments.

---

## [Decision Letter · Decision Letter 1]

4 Oct 2024

Normal male fertility in a mouse model of KPNA2 deficiency

PONE-D-24-18326R1

Dear Dr. Rother,

We’re pleased to inform you that your manuscript has been judged scientifically suitable for publication and will be formally accepted for publication once it meets all outstanding technical requirements.

Kind regards,

Xuejiang Guo, Ph.D.

Academic Editor

PLOS ONE

Additional Editor Comments (optional):

Reviewers' comments:

Reviewer's Responses to Questions

**Comments to the Author**

1. If the authors have adequately addressed your comments raised in a previous round of review and you feel that this manuscript is now acceptable for publication, you may indicate that here to bypass the “Comments to the Author” section, enter your conflict of interest statement in the “Confidential to Editor” section, and submit your "Accept" recommendation.

Reviewer #1: All comments have been addressed

2. Is the manuscript technically sound, and do the data support the conclusions?

Reviewer #1: Yes

3. Has the statistical analysis been performed appropriately and rigorously? 

Reviewer #1: Yes

4. Have the authors made all data underlying the findings in their manuscript fully available?

Reviewer #1: Yes

5. Is the manuscript presented in an intelligible fashion and written in standard English?

Reviewer #1: Yes

6. Review Comments to the Author

Reviewer #1: The authors have adressed all the suggestions and comments made in the revision. In the present version, all the raw data is completely available as a supporting information, which helps to understand the results. The information requiered in M&M and Discussion sections was included.

7. PLOS authors have the option to publish the peer review history of their article (what does this mean?). If published, this will include your full peer review and any attached files.

Reviewer #1: **Yes: **Geraldine Schlapp

---

## [Editor Report · Acceptance letter]

10 Oct 2024

PONE-D-24-18326R1 

PLOS ONE

Dear Dr. Rother, 

I'm pleased to inform you that your manuscript has been deemed suitable for publication in PLOS ONE. Congratulations! Your manuscript is now being handed over to our production team.

Kind regards, 

on behalf of

Prof. Xuejiang Guo 

Academic Editor

PLOS ONE